# Bone Regeneration in Critical-Sized Bone Defects Treated with Additively Manufactured Porous Metallic Biomaterials: The Effects of Inelastic Mechanical Properties

**DOI:** 10.3390/ma13081992

**Published:** 2020-04-24

**Authors:** Marianne Koolen, Saber Amin Yavari, Karel Lietaert, Ruben Wauthle, Amir A. Zadpoor, Harrie Weinans

**Affiliations:** 1Department of Orthopaedics, University Medical Centre Utrecht, 3584 CX Utrecht, The Netherlands; S.AminYavari@umcutrecht.nl (S.A.Y.); h.h.weinans@umcutrecht.nl (H.W.); 23D Systems Healthcare, 3D Systems Leuven, 3001 Leuven, Belgium; karellietaert@gmail.com (K.L.); ruben.wauthle@3dsystems.com (R.W.); 3Department of Biomechanical Engineering, Faculty of Mechanical, Maritime and Materials Engineering, Delft University of Technology, 2628 CN Delft, The Netherlands; a.a.zadpoor@tudelft.nl

**Keywords:** porous titanium, additively manufactured, biomaterials, bone regeneration

## Abstract

Additively manufactured (AM) porous metallic biomaterials, in general, and AM porous titanium, in particular, have recently emerged as promising candidates for bone substitution. The porous design of such materials allows for mimicking the elastic mechanical properties of native bone tissue and showed to be effective in improving bone regeneration. It is, however, not clear what role the other mechanical properties of the bulk material such as ductility play in the performance of such biomaterials. In this study, we compared the bone tissue regeneration performance of AM porous biomaterials made from the commonly used titanium alloy Ti6Al4V-ELI with that of commercially pure titanium (CP-Ti). CP-Ti was selected because of its high ductility as compared to Ti6Al4V-ELI. Critical-sized (6 mm diameter) femoral defects in rats were treated with implants made from both Ti6Al4V-ELI and CP-Ti. Bone regeneration was assessed up to 11 weeks using micro-CT scanning. The regenerated bone volume was assessed ex vivo followed by histology and biomechanical testing to assess osseointegration of the implants. The bony defects treated with AM CP-Ti implants generally showed higher volumes of regenerated bone as compared to those treated with AM Ti6Al4V-ELI. The torsional strength of the two titanium groups were similar however, and both considerably lower than those measured for intact bony tissue. These findings show the importance of material type and ductility of the bulk material in the ability for bone tissue regeneration of AM porous biomaterials.

## 1. Introduction

Treatment of substantial (critical size) bone defects that often result from removal of bone tumors or trauma is still a major challenge in orthopedic surgery. Multiple treatment strategies including autografts or allografts are currently being used. However, non-union is observed in 4.9% of the cases treated for bone fracture [1]. In 23% of ankle arthrodesis cases, for example, non-unions persist, requiring multiple invasive procedures and causing prolonged immobilization and even permanent morbidity [2]. It is therefore important to develop bone substitutes that stimulate bone tissue regeneration and help in overcoming bony non-unions.

Recent developments in free-form manufacturing techniques, such as advanced additive manufacturing (AM) technologies, have enabled fabrication of fully porous metallic biomaterials that could mimic the elastic mechanical properties of bone while offering unusually large surface to volume ratios [3,4,5,6,7]. In particular, the medical-grade titanium alloy Ti6Al4V-ELI processed with AM techniques such as direct metal printing has been extensively investigated as a potential bone substituting material during the last decade [8,9,10]. Although other materials such as pure titanium, tantalum, nitinol, and cobalt–chromium have been suggested as alternative materials for fabrication of AM bone substitutes, it is not clear what effects the material type has on the bone tissue regeneration performance of such biomaterials [11,12,13,14,15]. More specifically, the effects of the inelastic mechanical properties of the bulk material such as ductility on the bone tissue regeneration performance have been never studied before.

Commercially pure titanium (CP-Ti) and titanium alloy (Ti6Al4V-ELI) both have a modulus of elasticity in the range of 100 to 115 GPa, whereas bone has an elastic modulus up to 20 GPa for cortical bone and as low as < 1 GPa for cancellous bone [16,17,18]. Using highly porous designs, the elastic modulus of AM porous titanium (both Ti6Al4V-ELI and CP-Ti) could be reduced to the levels comparable with those of the native bone tissue [16,17,18,19]. There are, however, considerable differences between the inelastic mechanical properties of Ti6Al4V-ELI and CP-Ti. In particular, CP-Ti is much more ductile than Ti6Al4V-ELI with an ultimate elongation of ~30% as compared to 14% of elongation for Ti6Al4V-ELI [14]. Partially due to its higher ductility, AM CP-Ti has a much higher normalized (with respect to yield stress) fatigue strength as compared to Ti6Al4V-ELI [14]. The higher ductility of AM CP-Ti may result in more deformation at highly loaded locations and subsequently in a more uniform load distribution throughout the porous structure. This could, in turn, lead to higher volumes of de novo bone formation in the porous titanium scaffold.

We therefore hypothesized that that AM porous implants made from CP-Ti outperform their Ti6Al4V-ELI counterparts in terms of bone regeneration. We assessed the validity of this hypothesis using an animal model.

## 2. Materials and Methods

Sixteen male Wistar rats (Charles River, Sulzfeld, Germany) were accommodated in pairs of two under guidance in the animal facility of the University Medical Center Utrecht, The Netherlands. The research protocol was accepted by the animal ethics committee of the institution (105065-1/2014.III.12.105) and was in accordance with the national laws on animal research. Animals received standard food pellets and water ad libitum and were managed under controlled conditions (21 °C; 12 h light / 12 h darkness). At the age of 16 weeks and after 7 days of adjustment (weight = 340–370 g), a 6-mm critical size segmental bone defect was established in the right femur of each rat with a wire saw using a saw guide (RISystem, Davos, Switzerland) [20]. The surgeries were achieved aseptically under total anesthesia (1–3.5% isoflurane, AST Farma, Oudewater, The Netherlands). Briefly, the right hind leg was opened by a skin incision and dissection of soft tissue and division of underlying fascia. Using three proximal and three distal screws, a polyether ether ketone plate was attached to the femur. Subsequently, an AM implant made from CP-Ti (n = 8) or Ti6Al4V-ELI (n = 8) was implanted in the gap. The fascia and skin were closed using Vicryl Rapide 5-0 (Ethicon, Dülmen, Germany). Subcutaneous injection of pain medication (buprenorphine, 0.05 mg/kg body weight, AST Farma, Oudewater, The Netherlands) was given pre-operatively and twice a day for the following three days. Before the surgeries, the rats were given a single dose of antibiotics (enrofloxacin; 5 mg/kg body weight, Bayer, Mijdrecht, The Netherlands). The rats were euthanized after 12 weeks with an overkill of barbiturates (phenobarbital; 200 mg/kg body weight, TEVA Pharma, Haarlem, The Netherlands) and the femora were analyzed with micro-computed tomography (micro-CT), histology, and biomechanical testing.

AM porous CP-Ti implants were produced with direct metal printing (DMP, ProX DMP 320, 3D Systems Layerwise, Leuven, Belgium) using pure titanium powder conforming to ASTM Grade 1, while the powder used for construction of the AM Ti6Al4V-ELI implants conformed to ASTM B348 [21], grade 23. The chemical composition of both biomaterials included their typical elements, namely, oxygen, aluminum, and vanadium in Ti6Al4V-ELI and only oxygen in the case of CP-Ti. File preparation was done in the Magics software (version 24, Materialise, Leuven, Belgium) and DMP control software (3D Systems Layerwise, Leuven, Belgium). All implants were cylinders with a height of 6 mm, an outer diameter of 4 mm, and an inner diameter of 1 mm (leaving a central endosteal channel). The porous architecture was based on a dodecahedron unit cell [15]. The DMP machine used for fabrication of the implants enables controlling the oxygen level in the building atmosphere to under 50 ppm and is therefore very suitable for production of Ti alloys. The constructs were made on a CP-Ti build plate and cut from the plate by wire electrical discharge machining. Both CP-Ti and Ti6Al4V-ELI implants went through a post-production alkali–acid–heat surface treatment [22]. The compressive strength, elongation, and Young’s modulus were determined in an earlier study [14,23]. The surface morphologies of treated samples were evaluated with scanning electron microscopy (SEM) (JEOL, Tokyo, Japan).

For both types of implants, the pore size, strut size, pore volume, and porosity were detected with micro-CT scanning. To monitor the bone ingrowth, in vivo micro-CT scans of the femora were determined under general anesthesia (1–3.5% isoflurane) from all animals at 0, 4, 8, and 11 weeks after the surgery. The hind leg of the rat was fixed in the supine position to enable scanning the femur (scan time: 3 min; voxel size: 42 μm^3^; tube voltage: 90 kV; tube current: 180 A, Quantum FX; PerkinElmer, Waltham, MA, USA). From all datasets, bone volume inside (BVi, Figure 1) and outside (BVo, Figure 1) the 6-mm defect was determined using ImageJ (NIH, Redwood Shores, CA, USA). Global threshold values were chosen using visual inspection and were kept constant for all scans. Bone bridging was assessed on ex vivo scans with ImageJ. Bone bridging was quantified by measuring the shortest remaining gap size between the bones formed at the proximal and distal sides of the 6 mm defects.

Undecalcified histology was performed on two femora per group whose regenerated bone volumes (total BV after eleven weeks) were the closest to the means of their respective groups. The harvested femora chosen for histology were fixed in a 4% neutral formalin buffered solution for 1 week, dehydrated in ascending ethanol series (70–100%), and embedded in methyl methacrylate. Sections in the coronal plane of ~20 µm were obtained using a diamond saw (Leica SP1600, Leica Microsystems BV, Son, The Netherlands) and stained with 1% methylene blue (Sigma-Aldrich, Zwijndrecht, The Netherlands) and 0.3% basic fuchsin solution (Sigma-Aldrich, Zwijndrecht, The Netherlands) to stain bone pink and fibrous tissue blue. Serial sections (across the middle) were then evaluated for bone formation and bone–implant contact. All sections were visualized using an Olympus BX51 microscope (Olympus DP70 camera, Olympus, Hamburg, Germany).

The biomechanical stability of the grafted femora was assessed using a torsion test performed on the remaining six femora of each group to determine the maximum torque to failure and rotation at maximum torque. Both ends of each femur were fixed in a cold-cured epoxy resin (Technovit 4071, Heraeus Kulzer, Germany) after taking of the PEEK fixation plate. On the upper clamping side, a Cardan joint was applied to make sure that the samples were subjected to pure rotation without bending. The lower sides of the samples were simply secured. The tests were done until failure with a rotation rate of 0.5 s^−1^ using an electromechanical testing machine (ElectroPuls E10000, Instron, Norwood, MA, USA). Seven contralateral femora were added as controls, to compare our developed implants to native bone tissue. After harvesting the femora, soft tissues and PEEK plates were accurately removed. The samples were kept inside an in PBS-drowned napkin at −20 °C until further processing.

The data are given as means with standard deviation (SD) unless otherwise mentioned. In the analysis of the results of the micro-CT scanning, mixed-model analysis was used to test for statistical differences between both groups (i.e., CP-Ti and Ti6Al4V-ELI), with random intercept and correction for time (SPSS 22.0 software IBM, Armonk, NY, USA). Differences in gap size between both groups were analyzed with using the Student’s t-test. Assumptions of normal distribution were tested using a normality test and same variance using the Levene’s test. Differences in maximum torque and rotation at maximum torque between both groups and control femora were analyzed using one-way analysis of variation (ANOVA) and subsequent post hoc pairwise comparisons with Bonferroni adjustment. A p-value < 0.05 was considered statistically significant.

## 3. Results

All rats were capable of weight-bearing activities directly after surgery. The wounds healed without complications and all animals stayed healthy during the rest of the study. Animals had an average weight of 406 ± 33 g at time of implantation, which increased during follow-up with an average of 66 ± 17 g.

### 3.1. Porous Titanium Implants

Ti6Al4V-ELI implants had an average strut size of 210.5 ± 0.2 µm, an average pore size of 243.9 ± 0.4 µm, and a porosity of 79% (Table 1).

CP-Ti samples had an average strut size of 210.3 ± 3.8 µm, an average pore size of 244.5 ± 1.0 µm, and a porosity of 80%. The alkali–acid–heat treatment resulted in a titanium oxide layer with similar composition in terms of oxide and titanium with irregular nano-scale features (Table 1 and Figure 2) [22].

Macroscopic inspection and SEM analyses verified that CP-Ti looked more corroded and rough than the Ti6Al4V-ELI (Figure 2). As the elastic bulk mechanical properties were 103 GPa for CP-Ti vs. 113 GPa for Ti6Al4V-ELI, the apparent elastic properties for the two AM porous biomaterials are very similar, as was reported earlier [14]. Therefore, the most important difference between the two implants is a difference in plastic deformation (ductility) properties.

### 3.2. Micro-CT Analysis

There is a borderline significant difference between the groups (CP-Ti and Ti6Al4V-ELI) over time with a mean difference over time of 5.58 (CI −0.15–11.3, *p* = 0.055) for BVi, and no significant difference for BVo with a mean difference over time of 3.54 (CI −5.18–12.3, *p* = 0.399). At eleven weeks, CP-Ti showed more bone formation with BVi = 20.4 ± 7.6 mm^3^ and BVo = 18.0 ± 11.0 mm^3^ as compared to Ti6Al4V-ELI with BVi = 14.6 ± 7.2 mm^3^ and BVo = 12.1 ± 13.3 mm^3^ (Figure 3). There was no statistically significant (*p* = 0.369 BVi, *p* = 0.166 BVo) interaction between the time and material type, meaning that the effects are consistent over time.

### 3.3. Ex Vivo Micro-CT Analysis and Histology

Micro-CT scans over time visually showed that most bone was formed near the proximal sites of the titanium implants (Figure 4). Clear bone growth was also found inside the scaffold as well as in the medullary channel and somewhat outside the implants. At eleven weeks, no remodeling to the original bone cortical architecture was observed. In Ti6Al4V-ELI implants, there was some bone resorption at the distal ends. Histology confirmed more bone formation in the CP-Ti implants as compared to the implants from the Ti6Al4V-ELI group (Figure 5). The remaining gaps in the defects were filled with fibrous tissue. The architecture of the CP-Ti implants looked distorted with the implants somewhat broken into smaller pieces (Figure 5A), while the cellular structure of the Ti6Al4V-ELI implants remained intact until the endpoint (Figure 5C). None of the implants fractured. The bone formed around CP-Ti implants showed bigger lacunes filled with osteocytes (Figure 5B), whereas bone formed around the Ti6Al4V-ELI implants looked more compact (Figure 5D). In both groups, the newly formed bone was in close contact with the implant.

Complete bone bridging was observed in only in two of the defects treated with CP-Ti implants (Figure 6A). The average remaining gap size was slightly lower (not significant, *p* = 0.555) in CP-Ti implants (0.91 ± 1.70 mm) as compared to the Ti6Al4V-ELI implants (1.28 ± 1.34 mm) (Figure 7).

### 3.4. Biomechanical Testing

In the torsion test, only two Ti6Al4V-ELI specimens were tested as the other four fractured, indicating that they were very weak. The two samples with Ti6Al4V-ELI had an average maximum torque of 141 ± 175 N/mm, whereas the five samples with CP-Ti were within a narrower range of 100 ± 80 N/mm. These values were considerably lower than the average maximum torque measured for the intact control femora (442 ± 219 N/mm), with a significant difference between CP-Ti grafted femora and the intact femora (Figure 6B, *p* = 0.036). This was corroborated by the rotation data that showed a significantly higher rotation at the maximum torque for the CP-Ti group (34.4 ± 18.3 degrees) as compared to the intact femora (18.9 ± 6.8 degrees) (Figure 6C, *p* = 0.0096).

## 4. Discussion

The results of the current study show that the material type in general and the inelastic mechanical properties of AM porous biomaterials in particular play an important role in determining their bone tissue regeneration performance. The results also support the hypothesis of the study by demonstrating that the more ductile CP-Ti seems to exhibit improved bone tissue regeneration performance over T6Al4V-ELI implants, even though the elastic mechanical properties and topological designs of both types of implants are almost identical.

Both types of titanium implants were osteoconductive and showed no ectopic bone formation. CP-Ti is, however, weaker and exhibited some levels of distortion. This effect is, however, expected to disappear, as bone regeneration into the porous structure continues, particularly given the fact that bone tissue regeneration into implants could increase both the yield and fatigue strengths (by up to 7 times) [24].

In addition to more bone formation in the CP-Ti implants, CP-Ti implants also have other advantages over the Ti6Al4V-ELI implants. First, CP-Ti has no possibly hazardous or toxic alloying components such as V or Al [25,26,27]. Second, the high ductility that gives CP-Ti its relatively large plastic deformation could be suitable for certain applications such as intraoperative deformations used to match the patient-specific anatomy.

The elastic mechanical properties of both types of implants are in the range of those reported for the human cancellous bone and could therefore increase stresses of the ingrowing bone and minimize the stress-shielding phenomenon. The better bone formation of CP-Ti compared to Ti6Al4V-ELI could not only be explained by its better biocompatibility, but also by the higher ductile properties of CP-Ti. This more ductile behavior leads to large (plastic) deformations at highly stressed locations and thus damping these peak stresses and creating a more uniform stress distribution. The consequence of this might be a better mechanical stimulus for bone tissue regeneration within the scaffold of CP-Ti, as mechanical strain is considered to be one of the major factors determining the osteogenic behavior [28,29,30,31].

Another advantage of higher ductility is that localized stresses will lead to local deformation and not local failures. It has been recently shown that AM porous biomaterials experience local stress concentrations that originate from manufacturing imperfections [32]. Such stress concentrations lead to large (inelastic) deformations in more ductile materials such as CP-Ti, which would mean higher stimulus for bone tissue regeneration. In contrast, changes in stress concentration that occur because of local failures in more brittle materials, such as Ti6Al4V-ELI, are not helpful in terms of stimulating bone regeneration.

The mechanical strength of CP-Ti is, nevertheless, lower than that of Ti6Al4V-ELI. In dental implants, there have been reports of fractures of implants made from CP-Ti, although only in less than 1% of the cases [33,34,35,36]. These aspects should be taken into account when designing orthopedic implants based on CP-Ti [37].

Both CP-Ti and Ti6Al4V-ELI implants are bio-inert and do not degrade over time. They could therefore introduce a long-term risk of infection. The large surface area of such AM porous biomaterials could, however, be biofunctionalized using antibacterial coatings to prevent implant-associated infections [38,39,40].

Recent studies have clearly shown that the geometry of porous biomaterials that are used for bone tissue engineering and/or bone substitution plays an important role in terms of the cell response, the rate of bone regeneration, and consequently the fate of the biomaterials [41]. Therefore, it is important to also investigate the effects of curvature on the rate of tissue generation by comparing concave, convex, and planar surfaces.

Given the importance of topological design in adjusting the elastic mechanical properties of cellular structures [18], most of the current research is focused on optimizing the porous biomaterials. We show here for the first time that the inelastic mechanical properties of AM porous biomaterials could also be important for improving the bone tissue regeneration performance. This observation could open new avenues for research into bone substitutes based on CP-Ti.

## 5. Conclusions

We conclude that porous CP-Ti seems to have better bone tissue regeneration properties than Ti6Al4V-ELI, suggesting a revival of CP-Ti implants in particular for situations where implant strength is less crucial. The higher plastic deformation of CP-Ti can be used as an advantage for better bone regeneration into the porous structure, but also for creating a better fit of the porous (deformable) implant with the shape of the bone.

## Figures and Tables

**Figure 1 materials-13-01992-f001:**
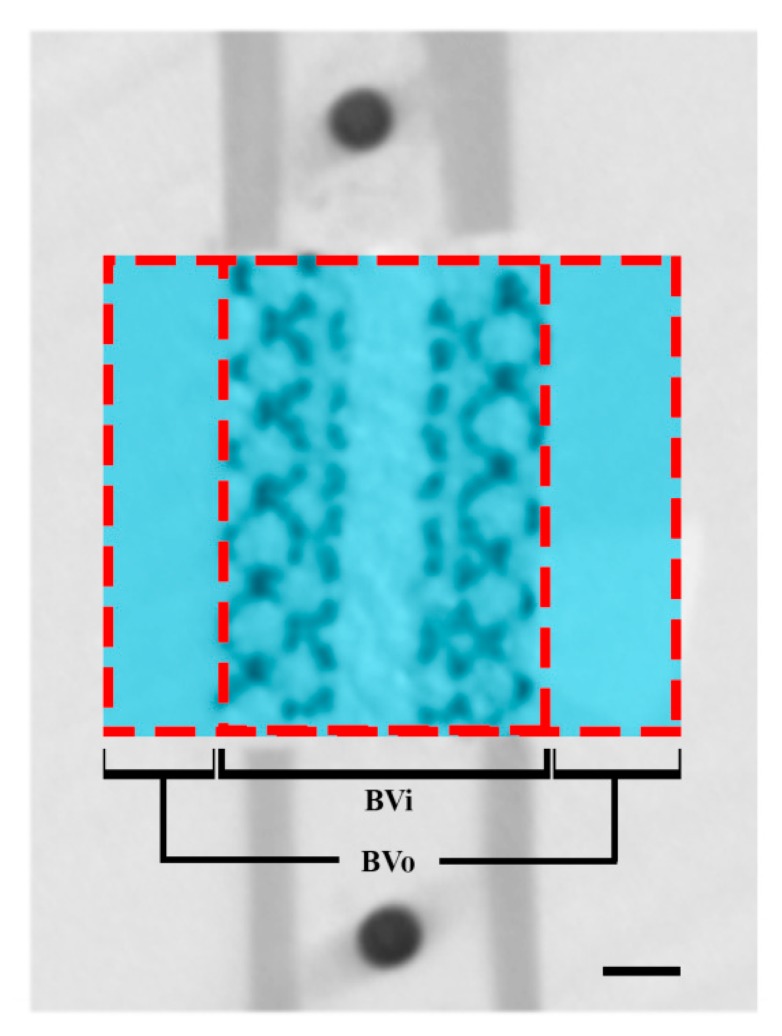
Region of interest on micro-CT scanning. Bone volume inside (BVi) and bone volume outside (BVo) are defined as bone formed inside and outside the porous space and the medullary canal of the titanium implants. Scale bar indicates 1 µm.

**Figure 2 materials-13-01992-f002:**
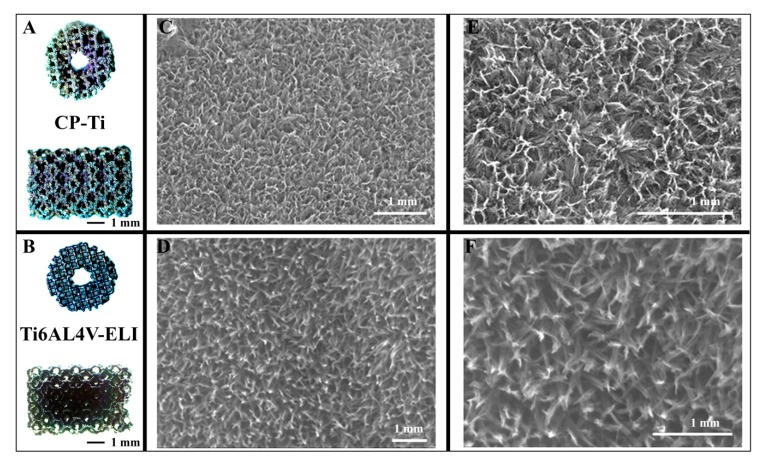
SEM images of CP-Ti and Ti6Al4V ELI porous implants. Macroscopic overview (**A**,**B**), and enlarged details (**C**–**F**) of the two different porous titanium implants. Panels (**E**,**F**) are zoomed in pictures of panels (**C**,**D**).

**Figure 3 materials-13-01992-f003:**
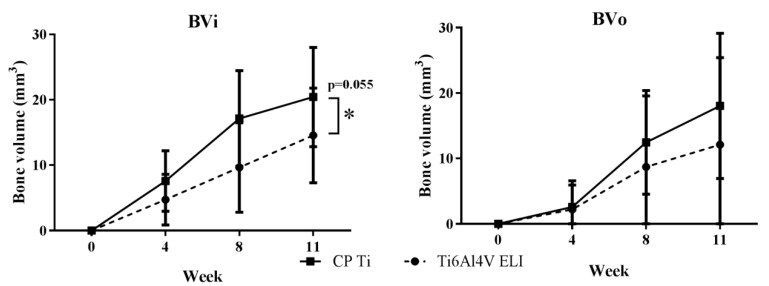
Longitudinal quantification of bone regeneration. In vivo µCT scans after zero, four, eight, and eleven weeks; BVi and BVo: defined as bone formed inside and outside the porous space and in the medullary canal of the titanium implants. Values are expressed as mean and SD (n = 8 for CP-Ti and n = 8 for Ti6Al4V-ELI). * Indicates (borderline) significant differences between groups.

**Figure 4 materials-13-01992-f004:**
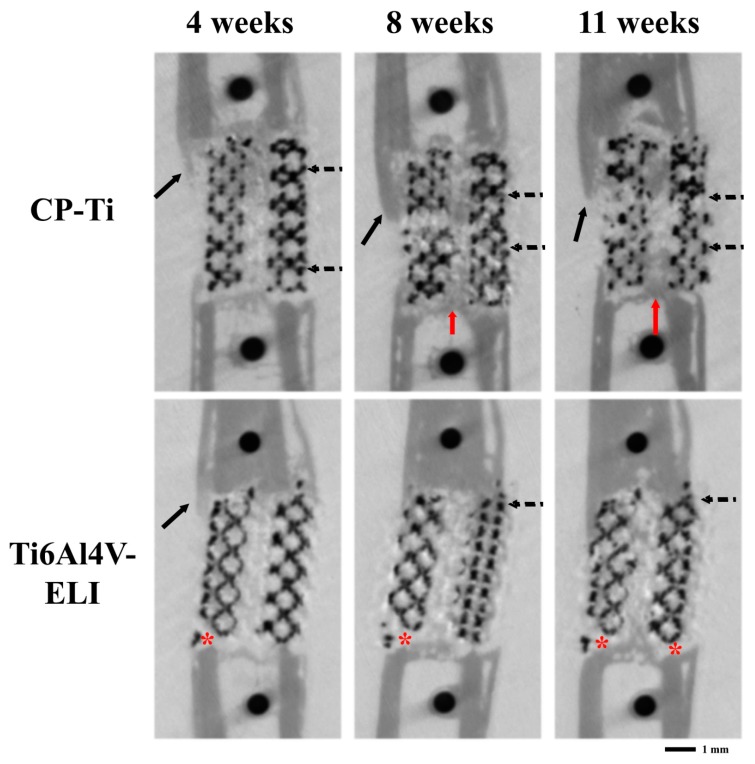
Representative longitudinal micro-CT scans of the femur illustrating the bone regeneration process. In vivo scans of defects grafted with two different titanium implants (CP-Ti or Ti6Al4V-ELI) after four, eight, and eleven weeks. In CP-Ti, there was more bone formation inside the canal of the titanium implants (red arrows), around (solid arrows), and inside (dotted arrows) the titanium implants compared to Ti6Al4V-ELI. Bone formation is mostly growing from the proximal ends. Distally from the Ti6Al4V-ELI implants, bone resorption is visualized (asterisk).

**Figure 5 materials-13-01992-f005:**
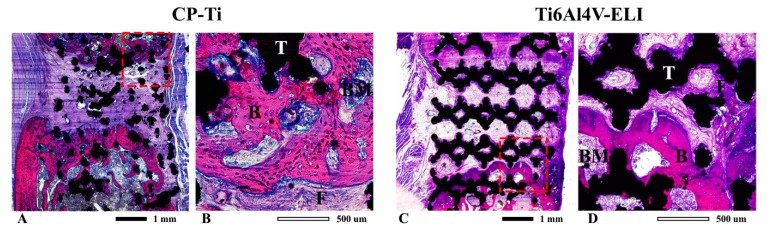
Histological evaluation of bone bridging. Representative transversal sections of femur defects eleven weeks after implantation of porous titanium implants; CP-Ti (**A**) and Ti6Al4V-ELI (**C**), including detailed view for CP-Ti (**B**) and Ti6Al4V-ELI (**D**). Sections are stained with basic fuchsine and methylene blue. Basic fuchsine stains bone pink, methylene blue stains fibrous tissue blue. B = bone, BM = bone marrow, F = fibrous tissue, T = titanium.

**Figure 6 materials-13-01992-f006:**
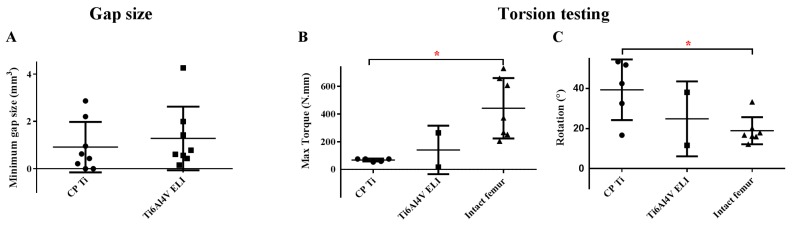
Bone bridging and mechanical strength. The remaining gap size after eleven weeks was used to indicate bridging success (**A**). Mechanical femoral strength (**B**) and rotation (**C**) after implantation of different types of porous titanium implants measured by torsion tests. * indicates significant differences between groups.

**Figure 7 materials-13-01992-f007:**
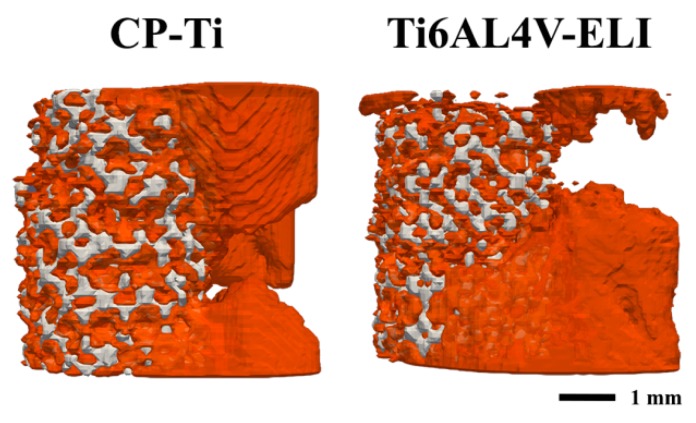
Illustration of bone bridging. Representative 3D micro-CT images showing the average extend of bone bridging of the CP-Ti and the Ti6Al4V-ELI titanium implants.

**Table 1 materials-13-01992-t001:** Properties of CP-Ti and Ti6Al4V-ELI porous titanium implants.

Parameter	CP-Ti	Ti6Al4V-ELI
Strut size	210.3 ± 3.8 µm	210.5 ± 0.2 µm
Pore size	244.5 ± 1.0 µm	243.9 ± 0.4 µm
Porosity	80 ± 0.5%	79 ± 0.3%
Pore volume	56.8 ± 0.3 mm^3^	56.0 ± 0.2 mm^3^
Surface area/volume	5.9 ± 0.1 µm^−1^	5.7± 0.0 µm^−1^
Elongation ($e_{max}$)	**	5.9%
Young’s modulus (E)	0.58 ± 0.02 GPa	0.55 ± 0.07 GPa
Compressive strength ($\sigma_{max}$)	**	19.4 ± 0.3 MPa
Surface composition	Oxygen 26.33% Titanium 73.67%	Oxygen 23.11% Titanium 69.3% Vanadium 2.64% Aluminium 4.95%

** Due to the ductile behavior of the porous CP Ti material, no maximum compressive stress ($\sigma_{max}$) and strain at maximum compressive stress ($e_{max}$) could be registered.

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
