# Peer review of "Bone Regeneration in Critical-Sized Bone Defects Treated with Additively Manufactured Porous Metallic Biomaterials: The Effects of Inelastic Mechanical Properties"

_materials, 2020, doi:10.3390/ma13081992_

Round 1
Reviewer 1 Report
This paper addresses the production of customized CP-Ti and Ti-6Al-4V ELI implants via additively manufactured (AM). Porous CP-Ti has better bone tissue regeneration properties than Ti6Al4V-ELI. The higher plastic deformation of CP-Ti can be an advantage for improving bone regeneration into the porous structure.
It is an excellent article, well written and to be published. However, some points remain to be clarified.
- The samples had a surface alkali-acid-heat surface treatment. How this treatment can play on surface topography and chemistry? Could also play a rule in the mechanical properties at interface bone-implant under load or torsion?
- Could the interaction of cells with the nano-topography of the surface and lead to modulation of mechano-transduction improved cell proliferation?
- It is not clear which is the difference observed from Porous CP-Ti has better bone tissue regeneration properties than Ti6Al4V-ELI?
- How the Al or V may have role to play after the alkali-acid-heat surface treatment?
- The position of figure 1 must be changed.
Reviewer 2 Report
In general, the work aim is clear, the methods are well described, and the writing style is very good. However, in order to accept the manuscript, the authors should consider and respond to some important points as below:
- The surgically-induced femoral defect, is it possible that this defect can be healed using metal work and without any materials? The authors should clarify this point in light of previous studies that used experimental models to compare the effectiveness of different materials in bone regeneration.
- In the first part of results, there is a limited description about why there was a difference between the two materials with regards to the ductility. The authors would need to explain the elongation difference and how was that measured.
- There is only significant borderline difference between both materials involve bone volume, while the other assays comparing the two materials were descriptive with no quantitative differences, therefore the conclusion about difference in bone generation is not adequately supported. The authors are advised to apply some quantification of their histology or micro Ct to further validate their hypothesis. Alternatively, the conclusions throughout the paper should be modified to show that no was no enough evidence for differences in regenerative support between the two materials apart from limited difference in the bone volume.
Reviewer 3 Report
PLEASE SEE THE UPLOADED FILE, "REVIEW-04-04-20".

Round 2
Reviewer 2 Report
The revised version and author response have addressed my comments.
Reviewer 3 Report
I am completely satisfied with the Revised Manuscript.